# Mechanism Study of Ultrasonic Vibration-Assisted Microgroove Forming of Precise Hot-Pressed Optical Glass

**DOI:** 10.3390/mi14071299

**Published:** 2023-06-24

**Authors:** Shengzhou Huang, Chengwei Jiang, Zhaowei Tian, Fanglin Xie, Bowen Ren, Yuanzhuo Tang, Jinjin Huang, Qingzhen Gao

**Affiliations:** 1School of Artificial Intelligence, Anhui Polytechnic University, Wuhu 241000, China; 2School of Mechanical Engineering, Anhui Polytechnic University, Wuhu 241000, China; 2210120117@stu.ahpu.edu.cn (C.J.); 15955817454@163.com (B.R.); t3375729@163.com (Y.T.);; 3Anhui East China Photoelectric Technology Research Institute, Wuhu 241002, China; 4Wuhu Changpeng Auto Parts Co., Ltd., Wuhu 241002, China

**Keywords:** optical components, ultra-precision modeling, ultrasonic vibration-assisted forming, finite element analysis

## Abstract

Microgroove structures with helical pitches in a wavelength level are increasingly required in optical areas. However, conventional manufacturing techniques generate relatively high stresses during pressing, resulting in poor precision when forming microgrooves. This paper reports on the mechanism of the ultrasonic vibration-assisted microgroove forming of precise hot-pressed optical glass. A finite element (FE) thermocompression model of the viscoelastic material was developed and the entire forming process was numerically simulated using coupled thermal-structural analysis. The analysis of several process parameters was carried out using orthogonal experiments, from which the optimum combination of parameters was selected. The glass thermoforming process is also assisted by ultrasonic vibration. The thermal and mechanical effects of vibration improved material flow and optimized forming results. The average maximum stress in the glass during the forming process was only 3.04 × 10^−3^ Mpa, while the maximum stress in the hot-pressing stage without ultrasound was 1.648 Mpa. The stress results showed that the material-forming stress is significantly reduced.

## 1. Introduction

In recent years, 3D micro surface structures have gained increasingly more important applications in the fields of optics, optoelectronics, and mechanical industries [1,2,3]. These optical elements with microstructures are widely used in optoelectronics, medical environments, and other fields, due to their unique optical properties and good mechanical properties [4,5,6,7]. As a typical binary micro-optical element, microgrooves have a feature size and period size smaller than the wavelength of light, and have good imaging properties and high-diffraction efficiency [7], which can be used as microfluidic channels or fluid switches in biochips. Otherwise, it can also be used to coat silicon plates to improve solar power generation efficiency in solar power generation.

High-efficiency pressing is an effective method to fabricate microgrooves. The traditional microgroove stamping method can produce microgrooves, but either the forming accuracy is low or the surface quality is poor. High residual stress leads to shape change after stamping due to the tendency of glass to adhere to the mold surface after cooling [8,9,10]. Since the surface quality of the finished product formed by the traditional preparation method is not very ideal, an auxiliary means is required to improve the surface quality of the molded element.

For this reason, several new composites forming processes such as ultrasonic vibration, electric field, laser, and other physical field-assisted forming have received extensive attention and have achieved rapid development. Among them, high-frequency/ultrasonic vibration-assisted forming has many advantages, such as reducing the plastic deformation resistance of the material, the friction force of the contact surface, improving the uniformity of the plastic deformation of the material, and suppressing the initiation and propagation of cracks [11,12,13,14,15,16]. To date, high-frequency and ultrasonic vibration-assisted forming has been well applied in forming processes such as wire drawing.

Ultrasonic-assisted plasticity forming technology began in the 1950s [17,18], when two Austrian scholars discovered during experiments that when high-frequency vibrations were applied, the deformation stress of the material was significantly reduced. Nowadays, ultrasonic vibration-assisted molding is widely used in many fields such as metals, ceramics, glass, polymers, composite welding, subtractive manufacturing, and plastic molding.

Tsai and Hung developed special equipment for ultrasonic vibration-forming glass. They used this equipment to perform thermocompression tests on microgrooves and Fresnel lenses of K-PSK100 glass. Experimental results revealed the great potential of ultrasonic vibration in fabricating high-precision microstructures [19,20,21]. Later on, Nguyen et al. studied the ultrasonic vibration-assisted hot-pressing process based on the ultrasonic thermal effect. Their experimental results showed that the high-energy ultrasonic vibration increases the temperature of the K-PSK100 glass, and the ultrasonic vibration exhibits a thermal effect. As a result, the molding pressure is significantly reduced, while the microscopic formability of the glass is significantly improved [20,21,22]. On the other hand, Tianfeng Zhou and Jiaqing Xie studied the mechanism of the ultrasonic-assisted glass molding process (UGMP) based on the ultrasonic mechanical effect. Their results showed that ultrasonic vibration can shorten the contact time between the resin preform and the mold, and make the stress distribution uniform within the formed microgrooves [11,20]. In addition, they also tested the mechanism of friction on glass (K-PG375) filling in UGMP, and proposed some methods to reduce friction and improve flow. The research results also confirmed the influence of the mechanical effect of ultrasonic vibration on glass formability, and briefly introduced the thermal effect of ultrasonic vibration.

At present, the theory on the effect of vibration on the interior and surface of plastically worked materials is based on two fundamental effects [23,24]. That is, the volume effect that affects the mechanical properties and the surface effect that affects the sliding friction. The volume effect is generally understood as the temperature of the material increases after the internal grains rotate. The thermal effect related to the crystal dislocation then softens, resulting in a decrease in the dynamic deformation resistance of the material. The surface effect is generally explained as the ultrasonic vibration on the mold and the workpiece. The influence of friction is manifested in reducing the stick–slip between the mold and the workpiece, improving the surface quality and geometric accuracy of the molded parts, reducing the wear consumption of the mold, and prolonging the tool life. The reason for the surface effect may be explained as follows. Firstly, the mold and the workpiece are instantaneously separated due to vibration, which is conducive to the entry of lubricating oil into the friction zone. Secondly, the friction vector in the vibration cycle is reversed, which is conducive to deformation processing. Lastly, the local thermal effect can improve the local adhesion phenomenon [25,26,27].

In general, there is no clear understanding of the macro and micro mechanisms of plastic deformation of materials under high-frequency dynamic loading [28]. At present, the use of ultrasonic-assisted fabrication of microstructures is a very common method, but in the field of hot pressing, resins and other low-melting-point materials are mainly used to prepare microstructures, and there are not many studies on glass materials. In addition, this research also uses the method of orthogonal experiment to explore the combination of process parameters, so as to find the optimal parameter combination.

In this paper, firstly, the viscoelastic model of glass at high temperature is established. Secondly, the finite element simulation of the interface contact between the glass and the mold is carried out. Then, the formation effect of the microgrooves under the pressing conditions without ultrasonic vibration and with ultrasonic vibration is studied, respectively. Finally, through the comparison between ultrasonic vibration-assisted forming and conventional forming, it is confirmed that the forming accuracy of the microgroove has been significantly improved. Moreover, the stress distribution of the glass has also been significantly improved with the assistance of ultrasonic vibration, and the adhesion between the glass and the surface of the microgroove is better.

## 2. Theoretical Study

### 2.1. Glass Viscoelastic Material Model

In general, any material will exhibit a combination of elasticity and viscosity. The physical properties of glass materials are highly dependent on temperature changes. It is a hard and brittle material at room temperature, and between its transition temperature and softening temperature, it exhibits some properties not found in other materials. At this temperature, it exhibits mixed properties of elastic solids and viscous fluids, with typical viscoelastic behavior: creep and relaxation [29]. In linear viscoelasticity theory, this property can be approximated by several types of spring-buffer mechanical models. Figure 1 is the most widely used generalized Maxwell model [30,31].

The generalized Maxwell model consists of several spring-damped systems connected in parallel. It provides a comprehensive characterization of the creep and stress relaxation properties of viscoelastic materials. The $g_{i}$ is the *i*-th branch in the model; $E_{i}$ is the Young’s modulus of the i-th spring; and $\eta_{i}$ is the viscosity of the i-th damper.

For viscoelastic materials, the characteristic expression of the shear stress σ(t) is as follows:(1)$$\sigma (t)=\int_{0}^{t}G(t-\tau )\frac{d\varepsilon (\tau )}{d\tau }d\tau +G(t)\varepsilon (0)$$
where G(t) is the shear stress relaxation modulus function, which is expressed in the form of Prony Series in Maxwells model as follows:(2)$$G(t)=G_{\infty }+\sum_{i=1}^{N}G_{i}\exp(\frac{-t}{\tau_{i}})$$
where $G_{i}$ is the shear modulus component, $\tau_{i}$ is the stress relaxation time component, $G_{\infty }$ is the shear modulus value for an infinite time, and $G(0)=G_{\infty }+\sum_{i=1}^{N}G_{i}$.

It is easy to know that the differential stress–strain relationship of the n-dimensional generalized Maxwell model can be expressed as follows:(3)$$\frac{d\varepsilon }{dt}=\frac{d\varepsilon_{i}}{dt}=\frac{d\sigma_{i}}{dt}\frac{1}{E_{i}}+\frac{\sigma_{i}}{\eta_{i}};\sigma =\sum_{i=0}^{n}\sigma_{i}$$

In this expression, ε and σ are the total strain and stress of the model, $\varepsilon_{i}$ and $\sigma_{i}$ is the strain and stress of the *i*-th model, $E_{i}$ and $\eta_{i}$ is the elastic modulus and viscosity of the *i*-th branch. Given a constant strain excitation, $\varepsilon_{0}$ as $\eta_{0}\to \infty$, the overall stress response of the model can be deduced from Equation (3) as
(4)$$\sigma =\sum_{i=0}^{n}E_{i}e^{-\frac{t}{\tau_{i}}}\varepsilon_{0}=E(t)\varepsilon_{0}$$
where $\tau_{i}=\frac{\eta_{i}}{E_{i}}$, E(t) represents the modulus of the time-dependent elastic modulus model, that is, the equivalent stiffness, which is usually expressed in the form of Prony series, as follows:(5)$$E(t)=E_{int}\sum_{i=0}^{n}g_{i}e^{-\frac{t}{\tau_{i}}}$$
where $E_{int}$ is the initial elastic modulus, and $g_{i}$ is the weight of the *i*-th branch, and the sum of its weights is 1, that is, $\sum_{i=0}^{n}g_{i}=1$.

The material used in this mold is tungsten carbide (WC). Due to its low-thermal expansion rate and high-temperature compressive strength, it is a relatively mature material in the field of glass molding [4]. The glass material is P-SK57 glass [32,33]. The material parameters of the two are shown in Table 1.

### 2.2. Glass Thermal Rheology Model

During compression molding, the relaxation time of glass at high temperature is closely related to temperature. When the temperature is low, the glass exhibits elastic properties similar to solids, its molecular motion is slow, the internal structure adjustment time is long, and the relaxation time is long. As the temperature gradually increases, the glass is mainly characterized by fluid viscosity, and the internal molecular motion speed is accelerated, the internal structure adjustment can be completed in a short time, and as a result, the corresponding relaxation time is short. Therefore, the same relaxation behavior can be achieved at a higher temperature and shorter time, or at a lower temperature and longer time. It can be seen that increasing the temperature or prolonging the relaxation time can affect the molecular motions equivalently. This property is called time–temperature equivalence, and viscoelastic materials with time–temperature equivalence are called simple thermo-rheological material.

Glass material is a typical simple thermo-rheological material. At different temperatures, the logarithmic curves of the relaxation modulus with time have the same shape but different degrees of shifts along the horizontal direction, as shown in Figure 2.

In the whole forming process, considering the viscoelastic time–temperature equivalent mechanical phenomenon of high-temperature glass, prolonging the relaxation time or increasing the temperature has an equivalent effect on the molecular motion. That is, it is equivalent to the viscoelastic mechanical behavior of the glass. This time–temperature equivalence can be described by the following equation:(6)$$G(T,t)=G(T_{r},\frac{t}{\alpha (T)})$$
where T is the actual temperature of the glass, $T_{r}$ is the reference temperature.

When the model is subjected to a differentiable time-varying strain, ε(t), according to the Boltzmann superposition principle, the total stress and total strain can be obtained. Considering the strong thermal sensitivity of glass, the equation is expressed as follows:(7)$$\sigma (t)=\int_{-\infty }^{t}E[(t-\xi ),T]\dot{\varepsilon }(\xi )d\xi =\int_{-\infty }^{t}E[\frac{\left(t-\xi \right)}{\alpha (T)},T_{r}]\dot{\varepsilon }(\xi )d\xi$$
where T represents the current temperature; $T_{r}$ represents the reference temperature; E is the stress relaxation modulus; α(T) is a displacement function related to temperature, the transfer factor is defined as follows:(8)$$\alpha (T)=\frac{\tau }{\tau_{0}}$$
where $\tau_{0}$ is the relaxation time at the reference temperature $T_{r}$, τ is the relaxation time at other temperatures T, and the transfer factor a is only a function of temperature.

It can be defined by the well-known Williams–Landel–Ferry (WLF) [13,14] equation as
(9)$$\log[\alpha (T)]=\log\frac{\tau_{i}(T)}{\tau_{i}(T_{r})}=\frac{-C_{1}(T-T_{r})}{C_{2}+(T-T_{r})}$$

In this expression, $\tau_{i}(T)$ represents the temperature-dependent relaxation time of branch *i*, $C_{1}$ and $C_{2}$ are fitting coefficients. In a wide temperature range, the heat-flow behavior of glass can be measured, analyzed, and measured, which is enough to simulate the traditional glass-forming process.

The WLF equation reflects the unique temperature dependence of the molecular chain motion of viscoelastic materials, and is suitable for fitting the stress-relaxation characteristics of viscoelastic materials such as glass in the temperature range of the glass transition temperature (*T_g_*) ~ *T_g_* + 100 °C. The stress relaxation phenomenon with too high of a temperature or too short of a relaxation time is difficult to realize experimentally. Firstly, the relaxation curve at the reference temperature can be obtained according to the test, and then the relaxation curve can be translated by using the WLF equation to obtain the relaxation curve that cannot pass the test.

## 3. Simulation Analysis

Figure 3 shows the process flow of microgroove formation. The forming process can be divided into four stages: heating, pressing and ultrasonic, annealing, and cooling, as shown in Figure 3a, b, c and d, respectively. First, the preform is placed on a lower mold, and the mold is used to heat the glass to tens of degrees Celsius above its transition point. Then, the upper mold is moved down to compress the glass, and ultrasonic vibrations are applied to the lower mold. The glass preform softens after heating, and is then pressed into shape by an upper mold. Finally, the pressure load is released, the mold release molded product is cooled to the mold release temperature, and taken out from the mold. In this simulation, a two-dimensional finite element model of seven micro-V-grooves is established in Abaqus software. Figure 4 is the two-dimensional view of the lower mold.

### 3.1. Optimal Combination Process Parameters without Ultrasonic Based on Orthogonal Experimental Method

The glass is affected by many factors during the hot-pressing process. In order to compare the influence of various process parameters on the molding quality and find the best combination of parameters, this study decided to use the method of orthogonal experimental design for simulation research.

In this orthogonal experiment, seven parameters including heating temperature (HT), pressing displacement (PD), hot-pressing time (HPT), compress time (CT), primary cooling time (PCT), secondary cooling time (SCT), and lower mold friction coefficient (LMFC) were selected to design the orthogonal experiment. In order to show the process sensitivity between the parameters, three different levels of seven process parameters were chosen to design the orthogonal experiments. Table 2 shows the horizontal factor table of the orthogonal experiment.

In this study, the stress and filling rate are important indicators of molding quality. Table 3 is the range result analysis of the displacement of glass in the groove in the orthogonal experiment results, where $K_{1}$, $K_{2}$ and $K_{3}$ are the simulation results, respectively, for the factor at the first, second, and third levels, ${\bar{K}}_{1}$, ${\bar{K}}_{2}$ and ${\bar{K}}_{3}$ are their average values.

Comparing the data in Table 3, the optimal levels of HT, PD, HPT, CT, PCT, SCT, and LMFC are 600 °C, 0.012 mm, 400 s, 250 s, 300 s, 100 s, and 0.17, respectively. Comparing the magnitude of the range, it is found that $R_{PD}$ > $R_{HT}$ > $R_{LMFC}$ > $R_{HPT}$ > $R_{PCT}$ > $R_{SCT}$ > $R_{CT}$, that is, the PD has the greatest impact on the glass-forming effect, followed by the heating temperature. However, the extreme difference analysis from the stress shows that the optimal levels of HT, PD, HPT, CT, PCT, SCT, and LMFC are 600 °C, 0.008 mm, 450 s, 300 s, 250 s, 90 s, and 0.15, respectively; the HT is more important. At last, the optimal combination was determined as 600 °C, 0.012 mm, 450 s, 250 s, 300 s, 100 s, and 0.15. In order to better represent this molding effect, the filling rate was used as a measure in this study. The filling rate is expressed by the ratio of the volume occupied by the glass in the lower mold microgroove ($V_{g}$) to the volume of the mold microgroove ($V_{m}$), namely:(10)$$Filling\;rate=\frac{V_{g}}{V_{m}}\times 100\%$$

Figure 5a is the maximum displacement cloud diagram, Figure 5b is the minimum stress cloud diagram, and Figure 5c is the optimal combination cloud diagram. Figure 6 shows the profile comparison of the final molding effect with different combinations of parameters. The best combination based on stress resulted in the lowest stress of 0.766 MPa, but its filling rate was only 78.88615%. The final selected parameter combination achieved a filling rate of 97.11919%, which is also not far from the best filling rate, and the morphological profiles basically overlap. However, the stress is only 1.648 MPa, which is 9.1% lower than the best filling rate result. In summary, the orthogonal experiment is indeed the optimal combination.

### 3.2. Influence of Ultrasonic Assistance on Surface Forming

According to the results of the orthogonal experiment, the optimal combination of this experiment is set for simulation. In order to explore the stress generated by ultrasonic vibration glass more carefully, this simulation only selects the stress nephogram of the glass during the hot-pressing step. Figure 7 shows the simulation results. Figure 7a is the stress cloud diagram of the glass hot pressed without ultrasonic vibration, and its maximum stress is 1.648 Mpa. Figure 7b is the stress cloud diagram of the glass after ultrasonic vibration is applied; its maximum stress is 1.04 × 10^−4^ Mpa.

In this experimental situation, the ultrasonic frequency, amplitude, and loading time are 30,000 Hz, 0.001 mm, and 400 s, respectively. In order to be closer to reality, the glass preform and the upper and lower molds are all set as deformable bodies in this simulation. It can be easily seen from Figure 7a,b that with the help of ultrasonic vibration, not only the stress distribution of the glass preform is improved, but also the deformation stress of the glass is greatly reduced. While the stress is greatly reduced after ultrasonic loading, the glass preform is more closely attached to the mold.

In order to see the optimization of glass forming more intuitively by ultrasonic vibration, Figure 8 is the maximum stress diagram of the glass obtained after loading the ultrasonic wave. As can be seen, the stress on the glass is greatly reduced compared to the optimal parameter combination with the aid of ultrasonic waves.

### 3.3. Influence of Ultrasonic Parameters on Surface Forming

In order to further observe the effect of ultrasonic assistance in the process of glass hot pressing, this experiment finally selects the three parameters of ultrasonic frequency (UF), amplitude (A), and loading time (LT) to design an orthogonal experiment, and performs a simulation according to the orthogonal experiment. Table 4 is the horizontal factor table of this orthogonal experiment.

Table 5 shows the range results of the simulation based on the three parameters of ultrasonic vibration. $K_{1}$, $K_{2}$, $K_{3}$ and $K_{4}$ represent the sum of the experimental indicators obtained from the orthogonal simulation experiment of each factor level in Table 4, and ${\bar{K}}_{1}$, ${\bar{K}}_{2}$, ${\bar{K}}_{3}$ and ${\bar{K}}_{4}$ are their average values. The following can be seen from the data in Table 5: $R_{LT}$ > $R_{UF}$ > $R_{A}$. That is, the time of ultrasonic loading has the greatest impact on stress, followed by frequency and amplitude.

## 4. Conclusions

This work mainly studied the simulations of the hot pressing of glass in detail with and without ultrasonic and loading ultrasonic vibration. The conclusions are summarized as follows:(1)The simulation results obtained by the optimal parameter combination show that the maximum stress of the glass is 1.648 Mpa, and the filling rate reaches 97.11919%. The result is better than any combination of parameters in the orthogonal experiment, and the combination obtained by the orthogonal experiment is indeed the best.(2)The average deformation displacement of the glass reaches 1947.561 nm, and the maximum filling rate is 97.37118%, while the glass is in the forming process. The average maximum stress during hot pressing is only 1.04 × 10^−4^ Mpa, which is a significant reduction in stress compared to the optimal combination of parameters in the orthogonal experiments. The results showed that the stress of the glass is greatly reduced with the help of ultrasonic vibration. In addition, the glass-filling rate was also better than that without the ultrasonic situation.(3)The thermal effect of ultrasonic vibration creates a softening effect in the glass, where the stiffness (resistance to deformation) of the glass is greatly reduced, allowing the glass to deform more under the same molding conditions.(4)In addition, the mechanical effect of ultrasonic vibration enhances the fluidity of the glass in the viscoelastic state, and the distribution of the glass on the mold is more uniform, which not only improves the stress distribution of the glass and reduces the stress of the glass, but also makes the molding effect better, and the filling rate in the V-groove is higher. That is to say, the surface shape of the final product fits better with the shape of the mold.(5)As shown in Table 5, in the ultrasonic sensitivity analysis with different frequency, amplitude, and loading time, it is found that the ultrasonic loading time has the greatest influence on the glass-forming stress.

## Figures and Tables

**Figure 1 micromachines-14-01299-f001:**
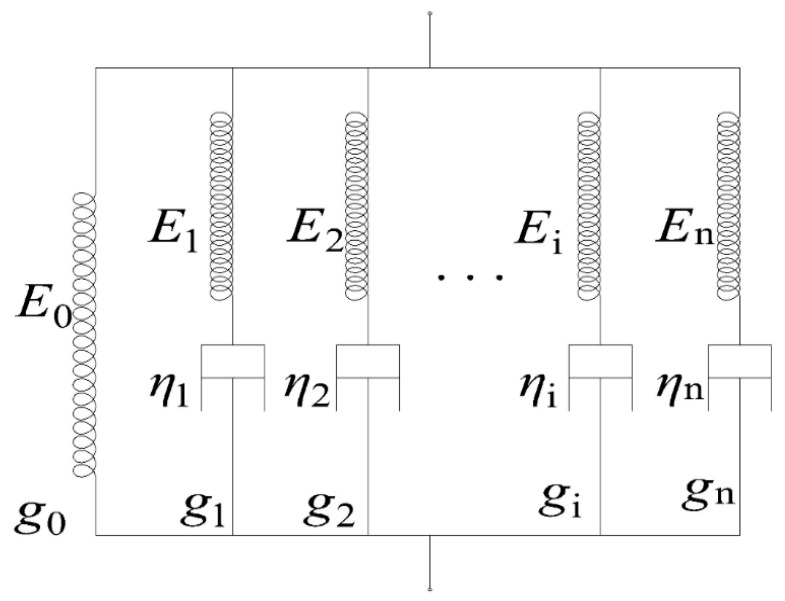
The generalized Maxwell model.

**Figure 2 micromachines-14-01299-f002:**
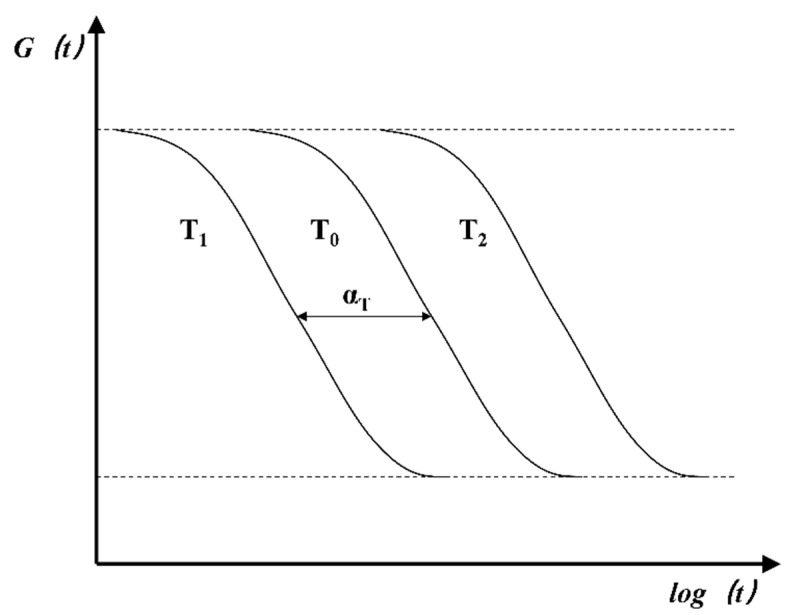
The glass thermo-rheological model.

**Figure 3 micromachines-14-01299-f003:**
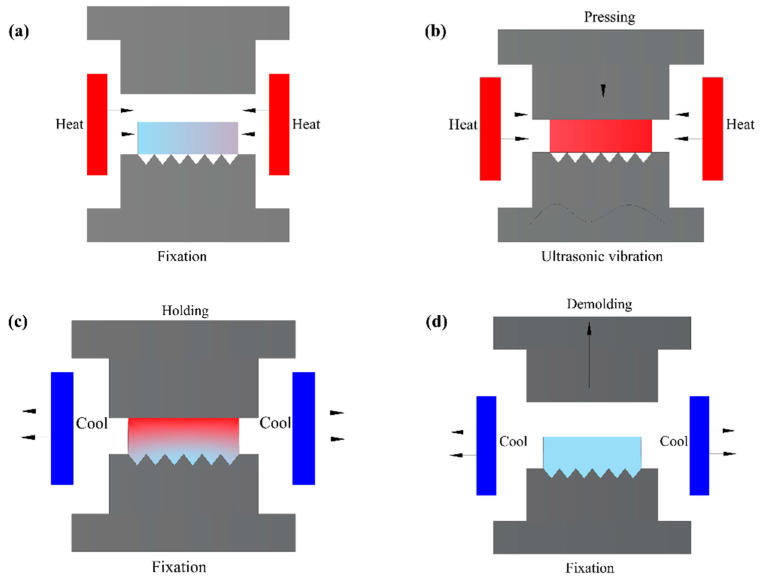
The flow chart of precise glass molding processing: (**a**) heating stage, (**b**) pressing and ultrasonic stage, (**c**) annealing stage, (**d**) cooling stage.

**Figure 4 micromachines-14-01299-f004:**
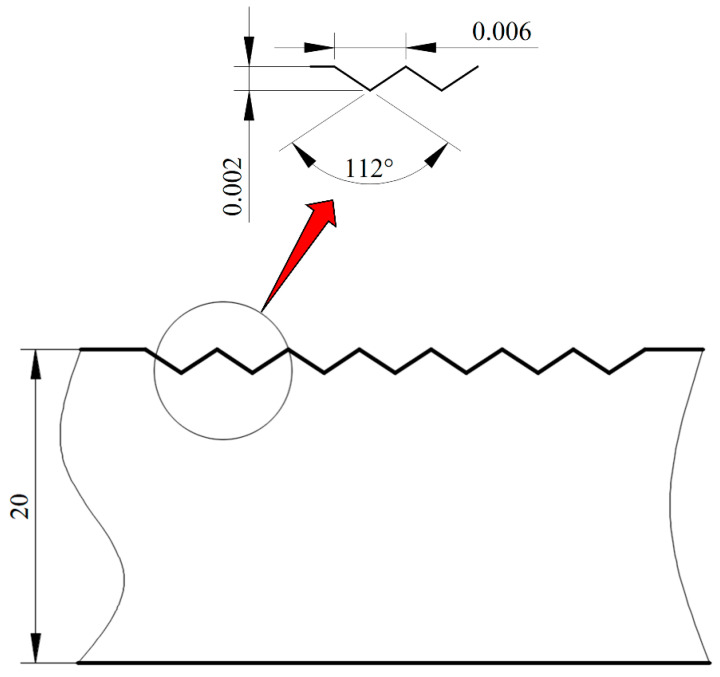
The lower mold size (unit: μm).

**Figure 5 micromachines-14-01299-f005:**
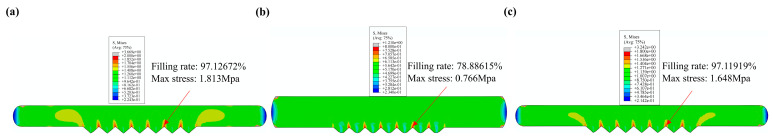
The comparison diagram of orthogonal experiment: (**a**) the maximum displacement, (**b**) the minimum stress, (**c**) the optimal combination.

**Figure 6 micromachines-14-01299-f006:**
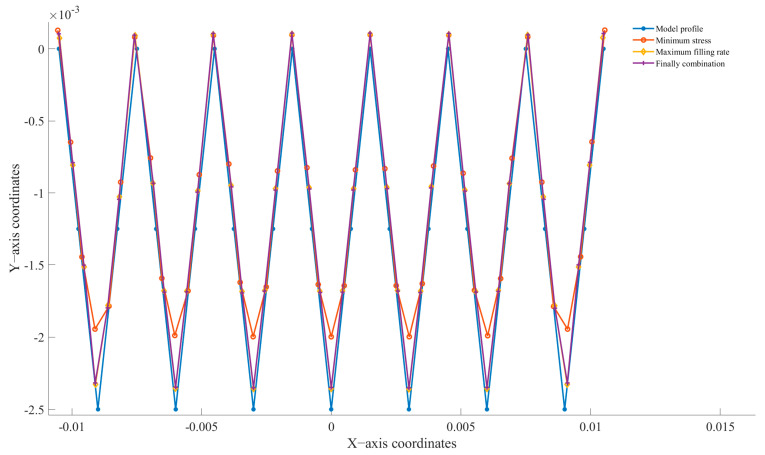
The combination of different parameters’ molding effect.

**Figure 7 micromachines-14-01299-f007:**
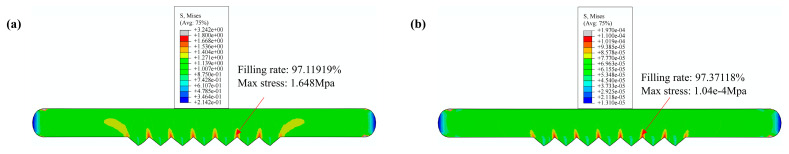
The cloud diagram of glass stress: (**a**) thermocompression stress cloud map of glass without ultrasonic, (**b**) thermocompression stress cloud map of glass with ultrasonic vibration.

**Figure 8 micromachines-14-01299-f008:**
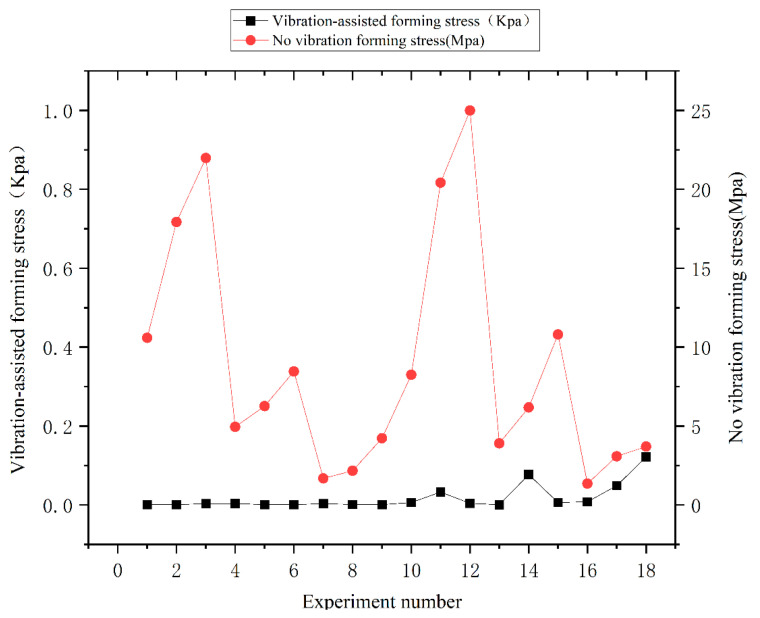
Comparison of maximum stress of glass with and without ultrasonic vibration.

**Table 1 micromachines-14-01299-t001:** Mechanical and thermal parameters of tungsten carbide mold (WC) and P-SK57 glass.

Material Attribute	Tungsten Carbide Mold	P-SK57 Glass
Young’s modulus of elasticity E (GPA)	710	93
Poisson’s ratio ν	0.22	0.249
Density ρ (kg/m^3^)	14,650	3010
Thermal conductivity K_c_ (w/m. °C)	63	1.1
Specific heat capacity Cp (j/kg·°C)	314	760
Coefficient of thermal expansion (/°C)	4.9 × 10^−6^	5 × 10^−6^
Conversion temperature (°C)	-	494

**Table 2 micromachines-14-01299-t002:** Table of three levels and seven factors of orthogonal experiment without ultrasonic vibration.

Factor Name	Heating Temperature (°C)	Pressing Displacement (mm)	Hot-Pressing Time (s)	Holding Time (s)	Primary Cooling Time (s)	Secondary Cooling Time (s)	Friction Coefficient of Lower Mold
Level 1	580	0.008	350	200	200	80	0.15
Level 2	590	0.01	400	250	250	90	0.17
Level 3	600	0.012	450	300	300	100	0.19

**Table 3 micromachines-14-01299-t003:** Range analysis of seven-factor orthogonal experiment.

Factor	Heating Temperature (°C)	Pressing Displacement (mm)	Hot-Pressing Time (s)	Holding Time (s)	Primary Cooling Time (s)	Secondary Cooling Time (s)	Friction Coefficient of Lower Mold
*K* _1_	10,549.5973	9471.731	10,553.4702	10,553.0023	10,552.4242	10,552.5476	10,551.0993
*K* _2_	10,553.9958	10,536.054	10,554.4337	10,554.2177	10,553.1137	10,553.0033	10,555.6882
*K* _3_	10,556.2817	11,652.0898	10,551.9709	10,552.6593	10,554.2784	10,554.3284	10,553.0873
${\bar{K}}_{1}$	1758.266	1578.622	1758.912	1758.834	1758.748	1758.758	1758.517
${\bar{K}}_{2}$	1759	1756.009	1759.072	1759.036	1758.852	1758.834	1759.281
${\bar{K}}_{3}$	1759.38	1942.016	1758.662	1758.777	1759.046	1759.055	1758.848
Range R	1.114	363.394	0.41	0.259	0.298	0.297	0.764
Sensitivity		Pressing displacement > heating temperature > friction coefficient of lower mold > hot-pressing time > primary cooling time > secondary cooling time > holding time					

**Table 4 micromachines-14-01299-t004:** Loading ultrasonic four-level three-factor orthogonal experiment table.

Factor Name	Frequency (Hz)	Amplitude (μm)	Loading Time (s)
Level 1	20,000	1	200
Level 2	25,000	2	300
Level 3	30,000	5	400

**Table 5 micromachines-14-01299-t005:** Range analysis of three-factor orthogonal experiment.

Factor	Frequency (Hz)	Amplitude (nm)	Loading Time (s)
*K* _1_	893.4145	893.8931	1674.317
*K* _2_	893.8633	893.4341	984.684
*K* _3_	893.9441	893.7873	577.08
*K* _4_	893.6181	893.7255	338.759
${\bar{K}}_{1}$	223.354	223.473	418.579
${\bar{K}}_{2}$	223.466	223.359	246.171
${\bar{K}}_{3}$	223.486	223.447	144.27
${\bar{K}}_{4}$	223.405	223.431	84.69
Range *R*	0.132	0.114	333.889
Sensitivity	Loading time > frequency > amplitude		

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
