# Peer review of "Mechanism Study of Ultrasonic Vibration-Assisted Microgroove Forming of Precise Hot-Pressed Optical Glass"

_micromachines, 2023, doi:10.3390/mi14071299_

Round 1
Reviewer 1 Report
The authors theoretical investigate the micro groove formation with ultrasonic vibration assisted glass thermoforming. An thermocompression model of the viscoelastic properties of glass materials are proposed and the assisted and non-assisted methods of glass thermoforming are compared. The results and discussion are well written and present enough new physics in order to be published. However, the manuscript requires improvements before it have my recommendation to be published in Micromachines.
I advise the authors to improve the following points on the new version of the manuscript:
1) In the introduction section, the authors write: "Ultrasonic-assisted plasticity forming technology began in the 1950s" a reference is required on this section.
2) Another section that requires references in the introduction section is: "In general, there is no clear understanding of the macro and micro mechanisms of plastic deformation of materials under high-frequency dynamic loading. At present, the use of ultrasonic-assisted fabrication of microstructures is a very common method, but in the field of hot pressing, resins and other low-melting-point materials are mainly used to prepare microstructures, and there are not many studies on glass materials."
3) The generalized Maxwell model is poorly described. There is no description of Figure 1, there is no definition of the parameters on Figure 1. The aproximations and assumptions are not described as well. The manuscript should be self contained and describe the model used in details.
4) What is E on equation 7?
5) Reference temperature is used as Tr in the manuscript. But in equation 8 it is changed. Please, fix it.
6) "It can be defined by the Williams–Landel–Ferry (WLF) [13,14], and the equation". Please fix this phrase. There are other typos and grammar mistakes on the manuscript. Improve the english in the new version of the manuscript.
7) The 3 levels on table 2 are not explained. Please, clarify that.
8) The explanation of K1, K2 and K3 should be improved. They are the sum of what specificaly?
In conclusion, the manuscript present enough new physics to deserve publication. However, it requires the above mentioned improvements in order to have my recommendation to be published in Micromachines.
The overall quality of the english is good. There are some small typos and grammar mistakes that should be addressed.
Author Response
Thank you for your letter and for the referee’s comments concerning our manuscript entitled “Mechanism study of ultrasonic vibration assisted micro groove forming of precise hot pressed optical glass”. Those comments are all valuable and very helpful for revising and improving our paper. We have studied comments carefully and have made corrections which we hope meet with approval.
1) In the introduction section, the authors write: "Ultrasonic-assisted plasticity forming technology began in the 1950s" a reference is required on this section.
The relevant references have been added to this section in the revised manuscript.
- Alers, G. A., (1955) Ultrasonic attenuation in zinc single crystals while undergoing plastic deformation. Physical Review, 97(4), 863. https://doi.org/10.1103/PhysRev.97.863
- Blaha, F., & Langenecker, B., (1959) Plastizitätsuntersuchungen von metallkristallen in ultraschallfeld. Acta Metallurgica, 7(2), 93-100. https://doi.org/10.1016/0001-6160(59)90114-2
2) Another section that requires references in the introduction section is: "In general, there is no clear understanding of the macro and micro mechanisms of plastic deformation of materials under high-frequency dynamic loading. At present, the use of ultrasonic-assisted fabrication of microstructures is a very common method, but in the field of hot pressing, resins and other low-melting-point materials are mainly used to prepare microstructures, and there are not many studies on glass materials."
The relevant references have been added to this section in the revised manuscript.
- Eaves, A. E., Smith, A. W., Waterhouse, W. J., & Sansome, D. H. (1975). Review of the application of ultrasonic vibrations to deforming metals. Ultrasonics, 13(4), 162-170. https://doi.org/10.1016/0041-624X(75)90085-2
- Xie, J., Zhou, T., Liu, Y., Kuriyagawa, T., Wang, X., (2016) Mechanism study on microgroove forming by ultrasonic vibration assisted hot pressing. Precis. Eng., 46, 270-277. https://doi.org/10.1016/j.precisioneng.2016.05.007
3) The generalized Maxwell model is poorly described. There is no description of Figure 1, there is no definition of the parameters on Figure 1. The aproximations and assumptions are not described as well. The manuscript should be self contained and describe the model used in details.
The Generalized Maxwell model is described in more detail in the manuscript and the parameters in the diagram are explained. The generalized Maxwell model consists of several spring-damped systems connected in parallel. It provides a comprehensive characterization of the creep and stress relaxation properties of viscoelastic materials. The gi is the i-th branch in the model; Ei is the Young's modulus of the i-th spring; and ηi is the viscosity of the i-th damper.
4) What is E on equation 7?
The E is the stress relaxation modulus on equation 7.
5) Reference temperature is used as Tr in the manuscript. But in equation 8 it is changed. Please, fix it.
Thank you for pointing out the error in the manuscript, which we have fixed in the manuscript.
6) "It can be defined by the Williams–Landel–Ferry (WLF) [13,14], and the equation". Please fix this phrase. There are other typos and grammar mistakes on the manuscript. Improve the english in the new version of the manuscript.
We have re-written the sentence to make sure that it is grammatically correct. We carefully checked the manuscript and corrected the grammatical errors in it.
7) The 3 levels on table 2 are not explained. Please, clarify that.
We have explained the three levels in the team table in the manuscript. In order to show the process sensitivity between the parameters, three different levels of seven process parameters were chosen to design the orthogonal experiments. The seven parameters in the orthogonal experiment table were analyzed in combination at three different levels.
8) The explanation of K1, K2 and K3 should be improved. They are the sum of what specificaly?
Thank you for your suggestion, we have explained below.
The K1 represents the sum of the simulation results obtained for the factor at the level 1; K2 represents the sum of the simulation results obtained for the factor at the level 2; K3 represents the sum of the simulation results obtained for the factor at the level 3.
9)The overall quality of the english is good. There are some small typos and grammar mistakes that should be addressed.
We carefully checked the manuscript and corrected the grammatical errors in it.
Reviewer 2 Report
Authors have studied the method to analyze the micro groove forming under the ultrasonic vibration. All contents of the manuscript describe the analytical model and FE simulation for studying the micro groove forming process. The simulation shows interesting and meaningful result, but it has not been validated with experiments. Authors should add the experiment section and verify the FE simulation with the experiment result.
Some of sentences have typos and grammar errors. Please check the manuscript.
Author Response
Thank you for your letter and for the referee’s comments concerning our manuscript entitled “Mechanism study of ultrasonic vibration assisted micro groove forming of precise hot pressed optical glass”. Those comments are all valuable and very helpful for revising and improving our paper. We have studied comments carefully and have made corrections which we hope meet with approval.
1)Authors have studied the method to analyze the micro groove forming under the ultrasonic vibration. All contents of the manuscript describe the analytical model and FE simulation for studying the micro groove forming process. The simulation shows interesting and meaningful result, but it has not been validated with experiments. Authors should add the experiment section and verify the FE simulation with the experiment result.
The experimental results are indeed more convincing evidence. Unfortunately, we do not currently have such experimental conditions. In future research, we intend to introduce experiments to validate our simulation results. Ultrasound-assisted forming is indeed a very useful forming method. However, our study can also be supported by the research of other scholars.
Wen, W. X., Li, L. Y., Li, Z., Ruan, W. Q., Ren, S., Zhang, Z. X., ... & Ma, J. (2023). Ultrasonic vibration-assisted multi-scale plastic forming of high-entropy alloys in milliseconds. Rare Metals, 42(4), 1146-1153. https://doi.org/10.1007/s12598-022-02171-2. In this study, the required pressure can be effectively reduced from 1.53GPa to 6.87MPa.
Liu, Y., Wang, C., & Bi, R. (2022). Acoustic residual softening and microstructure evolution of T2 copper foil in ultrasonic vibration assisted micro-tension. Materials Science and Engineering: A, 841, 143044. https://doi.org/10.1016/j.msea.2022.143044
In this study, it shows that the superimposed ultrasonic vibration can lead to grain refinement, resulting in homogeneous plastic deformation and flow stress reduction.
Xiao, M., & Jiang, F. (2022). Microstructural evolution of Fe-based amorphous alloy coatings via ultrasonic vibration-assisted laser cladding. Materials Letters, 322, 132520. https://doi.org/10.1016/j.matlet.2022.132520 In this study, it shows that the gradient structure formed with the aid of ultrasonic vibrations reduces the residual stresses in the coating and inhibits cracking.
shows that the superimposed ultrasonic vibration can lead to grain refinement, the decrease in LAGB fraction, KAM value and dislocation density, resulting in homogeneous plastic deformation and flow stress reduction.
2)Some of sentences have typos and grammar errors. Please check the manuscript.
We carefully checked the manuscript and corrected the grammatical errors in it.
Round 2
Reviewer 2 Report
Although the authors have not added the experiment result to validate the simulation, they suggested evidence to prove the FE simulation from relevant papers.
N/A